# Synthesis and Characterization of Lanthanide Metal Ion Complexes of New Polydentate Hydrazone Schiff Base Ligand

**DOI:** 10.3390/molecules27238390

**Published:** 2022-12-01

**Authors:** Izabela Pospieszna-Markiewicz, Marta A. Fik-Jaskółka, Zbigniew Hnatejko, Violetta Patroniak, Maciej Kubicki

**Affiliations:** Faculty of Chemistry, Adam Mickiewicz University, Uniwersytetu Poznańskiego 8, 61-614 Poznań, Poland

**Keywords:** Schiff base complexes, lanthanides, X-ray structures, spectroscopy, luminescence studies

## Abstract

The new homodinuclear complexes of the general formula [Ln_2_**L**_3_(NO_3_)_3_] (where **HL** is newly synthesized 2-((2-(benzoxazol-2-yl)-2-methylhydrazono)methyl)phenol and Ln = Sm^3+^ (**1**), Eu^3+^ (**2**), Tb^3+^ (**3a**, **3b**), Dy^3+^ (**4**), Ho^3+^ (**5**), Er^3+^ (**6**), Tm^3+^ (**7**), Yb^3+^ (**8**)), have been synthesized from the lanthanide(III) nitrates with the polydentate hydrazone Schiff base ligand. The flexibility of this unsymmetrical Schiff base ligand containing N_2_O binding moiety, attractive for lanthanide metal ions, allowed for a self-assembly of these complexes. The compounds were characterized by spectroscopic data (ESI-MS, IR, UV/Vis, luminescence) and by the X-ray structure determination of the single crystals, all of which appeared to be different solvents. The analytical data suggested 2:3 metal:ligand stoichiometry in these complexes, and this was further confirmed by the structural results. The metal cations are nine-coordinated, by nitrogen and oxygen donor atoms. The complexes are two-centered, with three oxygen atoms in bridging positions. There are two types of structures, differing by the sources of terminal (non-bridging) coordination centers (group A: two ligands, one nitro anion/one ligand, two nitro anions, group B: three ligands, three anions).

## 1. Introduction

For decades, the coordination chemistry of lanthanides has been a subject of great interest. Lanthanides and their complexes present attractive physical (e.g., spin crossover, single molecule magnetism, luminescence) [1,2,3,4,5,6,7,8,9,10,11,12,13,14], chemical [15], biological (antimicrobial, antiradical, anticancer) [16,17,18,19,20,21,22] and catalytic [23,24] properties. The versatility and robustness of lanthanides complexes with N, O donor ligands like hydrazones and their analogues caused the formation of the diverse classes of compounds.

Compounds containing polydentate hydrazone ligands exhibit a huge variety of molecular architectures. A flexible ligand may provide more possibilities of unique, thermally and kinetically stable, coordination modes with the lanthanide ions of various ionic radii. Lanthanide contraction and the electron configuration details, very important factors affecting the structures of the complexes, may be responsible for the difficulties in controlling the coordination environment around the metal ions. Additionally, lanthanides show the low stereochemical preference and the coordination compounds with these cations usually have high coordination numbers, e.g., typically 8 or 9 for the heavy lanthanides [25,26,27].

It is generally assumed that the complexes are formed in self-assembly process in which a disordered system of substrates is organized to supramolecular crystal structure. In such a process the role of solvent molecules, which can be and often really are the important parts of solid state structures, is also very important.

As a part of our ongoing research on the coordination properties of hydrazone ligands [28,29], we reported the study concerning the chelating abilities of the potentially polydentate new hydrazone Schiff base ligand (**HL**) 2-((2-(benzoxazol-2-yl)-2-methylhydrazono)methyl)phenol. A series of the lanthanide complexes of general formula [Ln_2_(**L**)_3_(NO_3_)_3_]·solvent (Ln = Sm^3+^ (**1**), Eu^3+^ (**2**), Tb^3+^ (**3a**, **3b**), Dy^3+^ (**4**), Ho^3+^ (**5**), Er^3+^ (**6**), Tm^3+^ (**7**), Yb^3+^ (**8**)) has been reported, and their structural and luminescent properties were studied in detail. Photoluminescence studies show that complexes Eu(**2**) and Tb(**3a**, **3b**) show characteristic emission in the visible region while Ho(**5**), Er(**6**), Tm(**7**) and Yb(**8**) in the near-infrared region. Photoluminescence spectra demonstrated that Er(**6**) and Yb(**8**) in these compounds can be effectively excited at 369 nm.

The results from this study show an interesting binding trend between N,N,O-donor ligand and selected Ln(III), as different coordination environments for the Ln atoms in the same molecule are observed. In the solid-state structures all the complexes are two-centered, lanthanide cations are always 9-coordinated in a distorted tricapped trigonal prism geometry, however, the metal coordination patterns are different depending on the crystallization conditions and/or metal cation. According to these differences, the complexes can be divided into two groups (**A** and **B**, Figure 1). The donor atoms of functional groups from the chelating ligands play here an important role. The coordination environments of both central cations in each complex are different and range from N_6_O_3_, through N_4_O_5_, N_2_O_7_ to O_9_ donor atom set.

## 2. Results and Discussion

### 2.1. Synthesis, ESI-MS and IR Spectroscopy, Thermal Analysis of the Complexes

The polydentate hydrazone Schiff base type metal ion complexes Sm^3+^ (**1**), Eu^3+^ (**2**), Tb^3+^ (**3a**), Tb^3+^ (**3b**), Dy^3+^ (**4**), Ho^3+^ (**5**), Er^3+^ (**6**), Tm^3+^ (**7**) and Yb^3+^ (**8**) containing ligand **HL** (Figure 2 and Appendix A) 2-((2-(benzoxazol-2-yl)-2-methylhydrazono)methyl)phenol were prepared.

Analytical data for the new compounds indicated that all complexes are in 2:3 metal to ligand stoichiometry and correspond to the general formula (in the solid state) [Ln_2_**L**_3_(NO_3_)_3_]·solvent. The metal ions are coordinated by organic ligands in a helical-like fashion, and two general coordination patterns can be identified (Figure 1). It needs to be noted, that deprotonation of the ligand **HL** is an important step to obtain products in good quality and yields.

The ligand **HL** was prepared from the condensation of semi product A and salicylaldehyde in molar ratio 1:1. The ligand structure was identified by elemental analysis, IR, ESI-MS and ^1^H, (Appendix A) ^13^C NMR and was further confirmed by X-ray diffraction of a single crystal. The complexes **1**–**8** were characterized by ESI-MS and IR spectral analysis. The ESI-MS of complexes showed mass fragmentation pattern which fits well with the formula suggested by elemental and X-ray analyses. The peaks corresponding to the free ligand are also present in the MS as a result of demetalation. The mass spectrum of ligand **HL** showed molecular ion peaks at *m*/*z* = 266 [C_15_H_13_N_3_O_2_-H]^−^, 268 [C_15_H_13_N_3_O_2_+H]^+^ which match the C_15_H_13_N_3_O_2_ ligand formula.

The IR spectra of the complexes provide some information regarding the bonding mode of the ligand **HL** and nitrate counterions. All IR spectra of complexes **1**–**8** exhibit three bands at 3322–3014 cm^−1^ region attributable to vibrations of >N-CH_3_ groups. The IR spectrum of ligand **HL** exhibits the bands at 3041 cm^−1^ and 2557 cm^−1^ characteristic of O-H stretching vibration and intramolecular OH∙∙∙N=C moiety, respectively. All IR spectra of compounds exhibit characteristic absorption band at around 1630 cm^−1^ attributed to C=N stretching mode confirming the Schiff base formation. The shift of this band to higher frequency, observed in the spectra of the complexes compared to the free ligand confirms the coordination of the nitrogen atom of azomethine group to the metal ions. The spectra of the complexes display vibrations indicative of coordinated nitrate groups. The *ν*(N–O) stretching frequencies are observed at 1475–1283 cm^−1^ region. The range of splitting confirms the bidentate coordinating behavior of the nitrate groups. The *ν*(C–O) phenolic frequency observed as a band at 1241 cm^−1^ for the ligand **HL**, is shifted in the complexes thus suggesting the participation of the oxygen atom of the deprotonated hydroxyl group in the formation of the M−O bonds. The metal complexes are also characterized by appearance of new bands at 519–501 cm^−1^ and 441–437 cm^−1^, which are assigned to *ν*(M–O) and *ν*(M–N) bending frequencies, respectively.

Thermogravimetric analysis was used to study the thermal decomposition and to confirm the melting point results. The thermal stability of representative complexes was investigated under a nitrogen atmosphere by thermogravimetric analysis (TGA) in the temperature range of 30–800 °C on the powder samples. The TG curves of representative compounds (Appendix A) showed that they exhibit very similar thermal decomposition processes. The DTA curves show a high exothermic peak corresponding to total decomposition of complex in one step, at 267–332 °C temperature range.

### 2.2. Crystal Structures

Figure 1 shows the perspective views of **HL**. The perspective views of the chosen representative molecules of complexes are shown in Figure 2 and Figure 3 and similar representations of all molecules are attached as Appendix A. Appendix A lists the relevant geometric characteristics of these molecules.

Basically, there are two structurally different types of the complexes. In both cases, the complexes are non-symmetric (with an exception of (Tb)**3b**, which lies on the three-fold symmetry axis in the quite rare cubic space group P2_1_3), of Ln_2_**L**_3_(NO_3_)_3_ composition, two-centered, triple bridged by ligand O_14_ oxygen atoms, with two lanthanide cations coordinated in the most typical way with the coordination number 9. However, the details are quite different. In the group **A** ((Sm)**1**, (Tb)**3a**, (Dy)**4** and (Er)**6**) one metal ion is coordinated (besides three bridging oxygen atoms, which come from three ligand molecules) by four nitrogen atoms from two ligand molecules and two oxygen atoms from one nitrate ion (N_4_O_5_), while the other—by two nitrogen atoms from the third ligand molecule, and four oxygen atoms from two nitrates, so it is overall N_2_O_7_. In the group **B** ((Eu)**2**, (Tb)**3b**, (Ho)**5,** (Tm)**7**, (Yb)**8**) one of the Ln ions is coordinated (again, besides bridging oxygens) solely by ligands (N_6_O_3_), and the second one only by nitrates (O_9_). It might be noted, that the structural differences are smaller in group **A**, with three out of four structures, **1**, **4**, and **6** closely isostructural (cf. Figure 4, the isostructurality indices are close to ideal values) than in group **B** (only (Tm)**7** and (Yb)**8**, i.e., two out of five, are isostructural).

In the Cambridge Structural Database [30] there are quite a lot examples of two-centered Ln complexes bridged by three oxygen atoms (1549 hits, of which 470 with both centers 9-coordinated), but we have found only few groups of complexes resembling relatively closely our new structures: tris(μ_2_-(E)-2-(2-pyridylmethyleneamino)phenolato-N,N′,O,O)-tris(nitrato-O,O′)-di-lanthanide(III) [31], tris(μ_2_-2-(2-hydroxypropyliminomethyl) phenoxo)-tris(nitrato-O,O′)-di-lanthanide(III) [32], and (μ-2,2′-[ethane-1,2-diylbis({[(pyridin-2-yl)methyl]azanediyl}methylene)]bis (6-formyl-4-methylphenolato))-(m-methoxo)-trinitrato-di- lanthanide (III) [33].

Such a complicated coordination uses the flexibility of the ligand. The molecule of the free **HL** is almost planar (dihedral angle between terminal ring systems is 11.21(6)º, cf. Appendix A, while all ligands in the complexes are significantly twisted. Such a conformation change allows the ligands for bridging two Ln ions, while at the same time for serving with two nitrogen atoms as coordination centers.

Each of the crystal structures of complexes contain varieties of different solvent molecules: acetonitrile, methanol, toluene, diethylether (Appendix A Crystal data). Interestingly, also in above listed examples of similar complexes, the crystal structures contained solvent molecules. This seems to suggest that the presence of appropriate—relatively small—solvent molecules is crucial for the possibility of obtaining single crystals, these molecules apparently help by filling the voids between the awkwardly shaped complex molecules. In general, weak C-H···N, C-H···O and C-H···π hydrogen bonds are predominant specific interactions which can be found in the structures.

### 2.3. Electronic Absorption Spectra and Luminescence Properties

The absorption spectrum of the Schiff base ligand **HL** (Figure 5) in acetonitrile showed three strong absorption bands in the range 220–450 nm (at about 239, 306 and 331 nm), that could be attributed to the π-π* or n-π* transitions [34,35,36]. The comparison of the absorption spectra of the ligand **HL** and the Ln^3+^ complexes in acetonitrile (Figure 5 and Appendix A) gave the numerical values of the maximum absorption wavelength and molar extinction coefficients (ε), which are listed in experimental data. Bathochromic shift of the absorption maxima and appearance of new bands were detected. These changes indicate formation of complexes between the lanthanide cations and ligand **HL**.

The intraconfigurational *f-f* transitions [37] are observed in the absorption and emission spectra of the Ln^3+^ ions. It should be noted that the ε values (molar absorption coefficient) of the forbidden *f-f* transitions are in general smaller than 10 dm^3^·mol^−1^·cm^−1^ [38].

Because of the strong ligand absorption, only a limited number of lanthanide ions show observable *f-f* transitions in the Schiff base complexes [39].

The Ho^3+^ absorption spectra (these ions are characterized by the highest values of molar absorption coefficients in the spectral range 500–800 nm) showed an increase in the absorption and bathochromic shift of the absorption bands, as presented in Figure 6 and Appendix A. The largest changes of the bands ^5^I_8_-^5^F_4_,^5^S_2_ (λ_max_~535 nm) and ^5^I_8_-^5^F_5_ (λ_max_~640 nm) in comparison with free HL are observed for (Ho)**5**. The changes of the absorption spectra are attributed to complexation of the ligand to the Ho^3+^ ion.

Photoluminescence study of the ligand **HL** and its corresponding complexes (Sm)**1,** (Eu)**2,** (Tb)**3a** and (Dy)**4** have been carried out in solution and solid state in the visible region. The exemplary excitation spectra of solid-state complexes from both analyzed groups are shown in Appendix A. For the ligand and complexes of group **A**, two bands located at about 270 and 370 nm are observed, while for the complexes of group **B** only one band is observed at 370 nm. The emission spectra of all solid samples display free ligand emission bands and are blue-shifted compared to the emission spectrum of the free ligand **HL** (Figure 7). The (Sm)**1**, (Tb)**3a** and (Dy)**4** (group **A** of complexes), at wavelength excitation λ_ex_ = 368 nm, exhibit broad emission bands with maximum at 453, 442 and 450 nm respectively, while (Tb)**3b** and (Eu)**2** (group **B**) two or three emission bands located at 410, 438 nm and 412, 440, 469 nm. The main emission peak of free **HL** ligand, in solid state is located at 512 nm for λ_ex_ = 370 nm. The very high intra ligand luminescence observed in the 400–650 nm spectral range, in the solid compounds, is indicative of nonefficient intramolecular energy transfer from the ligand to the excited states of Ln(III) ions and the lack of their bands in emission spectra.

The similarity in the excitation spectra of **HL** and **1**–**4** (Sm, Eu, Tb, Dy) complexes suggests that the absorption is ligand–based. The blue- or red-shifts with respect to **HL** may be ascribed to the coordination of Ln^3+^ ions with the ligand. The reduction of the emissions of these samples, in comparison with **HL**, was also observed.

Moreover, the luminescence emissions of the compounds **5**–**8** (Ho, Er, Tm, Yb) in the solid state were recorded at room temperature under 369 nm LED pumping from 900 to 1700 nm (Figure 8, Appendix A).

As shown in Figure 8, the luminescence spectrum of Ho^3+^ consists of three intense NIR bands which may be ascribed to the following transitions: ^5^F_5_-^5^I_7_ (966 nm), ^5^I_6_-^5^I_8_ (1178 nm) and ^5^F_5_-^5^I_6_ located at 1468 nm. The low intensity signals observed at about 1050 and 1300 nm are attributed to the ^5^S_2_-^5^I_6_ and ^5^S_2_-^5^I_5_ transitions observed in this ion. The peak position of ^2^F_5/2_-^2^F_7/2_ transition in Yb^3+^ is centered at 967 nm (Appendix A). The Tm^3+^ ion in **7** exhibits near-infrared and mid-infrared emissions at 795 nm (^3^H_4_-^3^H_6_) and 1450 nm (^3^H_4_-^3^F_4_) when **6** show a main band Er^3+^ with maximum at 1509 nm, which corresponds to the ^4^I_13/2_-^4^I_15/2_ transition. The intense photoluminescence in the near-IR region was observed for Er(**6**) and Yb(**8**) complexes.

The luminescence spectra of newly synthesized complexes **1**–**4** (Sm, Eu, Tb, Dy) were also measured in acetonitrile solution, Figure 9. The luminescence properties of (Sm)**1** and (Dy)**4** in solutions are similar to those in the solid state. Additionally, the complexes from group A display broad emission bands located around 440 nm, previously observed in the emission spectra of solid samples, and absence of *f-f* emission bands of Sm^3+^ and Dy^3+^ ions. In contrast, the ligand emission is attenuated in the spectrum of complex 2. Moreover, in this spectrum there are two weak bands related to the electronic *f-f* transitions in the Eu^3+^ ion (Figure 9a), i.e., transitions ^5^D_0_-^7^F_1_ and ^5^D_0_-^7^F_2_.

The observed differences in the emission spectra of the solid complexes Tb(III) of two groups) **3a** and **3b** were also reflected in the spectra of acetonitrile solutions (Figure 9b). In solution of compound **3a**, in addition to the ligand band located at about 440 nm, on its slope four bands reflecting the presence of Tb(III) ions, were also observed. These bands are at 491, 545, 585 and 623 nm and correspond to the ^5^D_4_-^7^F_j_ i.e., ^5^D_4_-^7^F_6_, ^5^D_4_-^7^F_5_, ^5^D_4_-^7^F_4_ and ^5^D_4_-^7^F_3_ transitions, respectively. Of them, the most intense transition is the first one, ^5^D_4_-^7^F_5_. Unlike **3a**, the spectrum of **3b** displays weak emission band corresponding to ^5^D_4_-^7^F_j_ transition in Tb^3+^ ion.

The hydrazone Schiff base ligand transfers the excitation energy to the Eu^3+^ and Tb^3+^ ions. The observed ligand-centered π-π* emission band at about 440 nm which indicates that the LMET in these complexes is not complete. The low intensity of the emission bands in **2** and **3b** (group complexes) proves that this process is inefficient. This process is more efficient for compound **3a**, in which Schiff base ligands are attached to both Tb^3+^ ions.

## 3. Materials and Methods

### 3.1. Materials

2-(1-methylhydrazinyl)benzoxazole was prepared according to the procedure described in [29]. Salicylaldehyde, 2-chlorobenzoxazole, methylhydrazine and Sm(NO_3_)_3_·6H_2_O, Eu(NO_3_)_3_·5H_2_O, Tb(NO_3_)_3_·6H_2_O, Dy(NO_3_)_3_·5H_2_O, Ho(NO_3_)_3_·5H_2_O, Er(NO_3_)_3_·5H_2_O Tm(NO_3_)_3_·6H_2_O, Yb(NO_3_)_3_·5H_2_O were used as received from Sigma-Aldrich.

### 3.2. Physical Measurements

The compounds were characterized using microanalyses (CHN), IR, ESI-MS, UV/Vis, ^1^H NMR, ^13^C NMR, and single crystal X-ray structural analysis.

IR spectra were recorded in the range of 4000–400 cm^−1^. Powders of sample were analyzed on a Nicolet iS50 FT-IR spectrometer, ATR technique was used. Mass spectra were recorded using electrospray ionization (ESI) techniques. Electrospray mass spectra were determined in methanol using a Waters Micromass ZQ spectrometer. Microanalyses (CHN) was obtained using a Perkin-Elmer 2400 CHN micro analyzer. NMR spectra were recorded in dimethylsulfoxide-d_6_ on a Bruker Ultrashield 300 spectrometer model operating at 300 MHz with chemical shifts (ppm) referenced to the deuterated solvent. NMR solvent was purchased from Deutero GmbH. Melting points were determined with a EZ-MeltA automated Melting Point Apparatus, Stanford Research Systems product, apparatus and are uncorrected. Thermogravimetric analyses (TGA) were carried out on a Setsys 1200 Setaran thermogravimetry under nitrogen atmosphere with a heating rate of 5 °C/min. Ultraviolet–visible (UV–Vis) spectra of the compounds in acetonitrile (at concentration of 2 × 10^−5^ M) were measured using Shimadzu UV PC 2401 spectrophotometer. Luminescence spectra in Vis range were recorded on a Hitachi F7000 spectrofluorometer at room temperature with a 1 cm quartz cell and filters 320 or 395, while in the NIR range were recorded by using an Andor Shamrock 500 spectrometer (300 L/mm–blaze 1200 nm) equipped with CCD camera iDus 420 and spectrometer QuantaMasterTM 40s (Photon Technology Instrumental, Birmingham, AL, USA) equipped with Photomultiplier Tubes H10330C-75 (800–1800 nm). The samples were excited by the use a LED revolver with high-power 360–370 nm LED.

### 3.3. Synthesis of the 2-((2-(benzoxazol-2-yl)-2-methylhydrazono)methyl)phenol, HL=C_15_H_13_N_3_O_2_

Ligand HL was synthesized within two subsequent steps starting from commercially available 2-chlorobenzoxazole according to the Appendix A. Formation of product thus formed was established by 1H NMR, 13C NMR, IR, ESI-MS spectroscopy and the X-ray diffraction methods. Ligand HL was synthesized according to the protocol described below.

To the colorless solution of **A** [29] (400.3 mg, 2.2 mmol) in EtOH_abs_ (8 mL) an equimolar amount of salicylaldehyde (268.7 mg, 234.5 μL, 2.2 mmol) was added. The mixture was allowed to react for 20 h under inert atmosphere in reflux and for another 4 h at room temperature. The white precipitate was observed. The product was filter under reduced pressure. The white needle shaped monocrystals suitable for the X-ray analysis were obtained by the slow evaporation of CH_3_CN. Yield: 513.8 mg, 1.9 mmol, 78.3%. Anal.: Calcd. for C_15_H_13_N_3_O_2_ (267.28 g mol^−1^): C, 67.40; H, 4.90; N, 15.72. Found C, 67.86; H, 4.87; N, 15.67%. Melting point: 182 °C. ^1^H NMR δ_H_(300 MHz, DMSO-d_6_) 10.86 (s, 1H), 8.26 (s, 1H), 7.65 (d, *J* = 7.8 Hz, 1H), 7.58 (d, *J* = 7.7 Hz, 1H), 7.48 (d, *J* = 7.7 Hz, 1H), 7.27 (dd, *J* = 15.6, 7.9 Hz, 2H), 7.16 (t, *J* = 7.2 Hz, 1H), 6.94 (dd, *J* = 7.4, 5.6 Hz, 2H), 3.69 (s, 3H). ^13^C NMR (DMSO-d_6_, 300 MHz): δ (ppm) 159.81; 156.61; 148.94; 141.97; 140.86; 139.75; 130.83; 128.95; 124.46; 121.86; 119.45; 117.03; 116.38; 109,62; 32.38. ESI-MS: *m/z* = 266 [C_15_H_13_N_3_O_2_-H]^−^, 267 [C_15_H_13_N_3_O_2_+H]^+^. IR (ATR, cm^−1^): ν = 3041 w (OH), 2575 w (OH····N), 1630 s (C=N), 1582 s (C=C), 1241 s (C–O) cm^−1^. UV-Vis (CH_3_CN): λ_max_/nm (ε/dm^3^ mol^−1^ cm^−1^) 239.5 (1.9 × 10^4^), 306.5 (2.5 × 10^4^), 331.0 (3.0 × 10^4^).

### 3.4. Synthesis of the Complexes. General Procedures

All complexes were prepared under similar conditions. To the colorless solution of ligand HL (20 mg, 40 μmol) in CH_3_OH (5 mL) the solution of appropriate lanthanide(III) nitrate salt [(60 µmol: 26.7 mg Sm(NO_3_)_3_·6H_2_O for **1**, 25.7 mg Eu(NO_3_)_3_·5H_2_O for **2**, 27.2 mg Tb(NO_3_)_3_·6H_2_O for **3**, 26.3 mg Dy(NO_3_)_3_·5H_2_O for **4**, 26.5 mg Ho(NO_3_)_3_·5H_2_O for **5**, 26.6 mg Er(NO_3_)_3_·5H_2_O for **6**, 26.7 mg Tm(NO_3_)_3_·6H_2_O for **7**, 26.9 mg Yb(NO_3_)_3_·5H_2_O for **8**] in CH_3_CN (5 mL) was added. No color change was observed. After 1 h the mixtures were treated with Et_3_N (5.6 µL, 40 µmol), which resulted in formation of clear yellow solutions. The final mixture was stirred at room temperature for 48 h under normal atmosphere. The solution volume was then reduced to 5 mL by roto-evaporation. Precipitation was carried out by addition of diethyl ether/methanol (1:1.5 mL). The solids were filtered off and dried in air. For the purpose of elemental analysis, the precipitates were dried under vacuum for 72 h in order to remove the solvent molecules [40,41]. The complexes obtained are microcrystalline variously colored powders, whose melting points (decomposition) are higher than that of the free ligand. Single crystals suitable for X-ray diffraction analysis were formed by slow diffusion of toluene (group **A**—**1**, **3a**, **4**, **6**) and diisopropyl ether (group **B**—**2**, **3b**, **5**, **6**, **7**) into sample to obtain by recrystallization of the solid from a minimum volume of CH_3_OH/CH_3_CN (1:1) at 4 °C over a period of 6–8 weeks. Single crystals were filtered under reduced pressure and air dried. The complexes were able to turn the solvent molecules into the crystal by recrystallization from mixed solvents. Solvents used in the processes of crystallization lead to the formation of different crystal structures and compositions, and thus affect their properties. Observed structural changes are characterized by a slight modification of the number of non-coordinated solvent molecules [42,43,44]. For the luminescence and UV-Vis studies, single crystals of the product were used [41].

***(1)*** 
*[Sm_2_(C_15_H_12_N_3_O_2_)_3_(NO_3_)_3_]·CH_3_CN **group A***


Light yellow precipitate, Yield: 24.36 mg, 0.0184 mmol, 63%. Analytical data Calculated for C_45_H_36_N_12_O_15_Sm_2_ (1285.57 g mol^−1^): *Calc.* C, 42.04; H, 2.82; N, 13.07. *Found:* C, 42.18; H, 2.84; N, 13.11%. Melting point: 324 °C (decomp.). ESI-MS: *m*/*z* = 684 [Sm(C_15_H_12_N_3_O_2_)_2_]^+^, 604 [SmC_15_H_12_N_3_O_2_(NO_3_)_3_]^−^, 268 [C_15_H_13_N_3_O_2_+H]^+^. IR (ATR, cm^−1^): ν = 3432 w (OH), 3058 w, 2936 w (N-CH_3_), 1639 s (C=N), 1603 m (C=C), 1460–1287 s, 835 m (NO_3_^−^), 1253 m (C–O), 1028 m (N–N) 503 w (Sm–O), 476 w (Sm–N) cm^−1^. UV-Vis (CH_3_CN): λ_max_/nm (ε/dm^3^ mol^−1^ cm^−1^) 308.4 (4.0 × 10^4^), 351.2 (2.6 × 10^4^).

***(2)*** 
*[Eu_2_(C_15_H_12_N_3_O_2_)_3_(NO_3_)_3_]·CH_3_CN·[(CH₃)₂CH]₂O **group B***


Orange precipitate, Yield: 21.53 mg, 0.0167 mmol, 58%. Analytical data Calculated for C_45_H_36_N_12_O_15_Eu_2_ (1288.78 g mol^−1^): *Calc.* C, 41.94; H, 2.82; N, 13.04. *Found:* C, 41.82; H, 2.97; N, 13.09%. Melting point: 308 °C (decomp.). ESI-MS: *m*/*z* = 683 [Eu(C_15_H_12_N_3_O_2_)_2_]^+^, 605 [EuC_15_H_12_N_3_O_2_(NO_3_)_3_]^−^, 268 [C_15_H_13_N_3_O_2_+H]^+^. IR (ATR, cm^−1^): ν = 3416 w (OH), 3108 w, 2975 w (N-CH_3_), 1642 s (C=N), 1603 m (C=C), 1457–1294 s, 831 m (NO_3_^−^), 1247 m (C–O), 1032 m (N–N) 505 w (Eu–O), 439 w (Eu–N) cm^−1^. UV-Vis (CH_3_CN): λ_max_/nm (ε/dm^3^ mol^−1^ cm^−1^) 307.6 (3.5 × 10^4^), 349.6 (2.2 × 10^4^).

***(3a)*** 
*[Tb_2_(C_15_H_12_N_3_O_2_)_3_(NO_3_)_3_]·CH_3_OH·CH_3_CN·C_6_H_5_CH_3_
**group A***
***(3b)*** 
*[Tb_2_(C_15_H_12_N_3_O_2_)_3_(NO_3_)_3_]·CH_3_CN·(C_2_H_5_)_3_NHNO_3_
**group B***


Light yellow precipitate, Yield: 21.18 mg, 0.0162 mmol, 54%. Analytical data Calculated for C_45_H_36_N_12_O_15_Tb_2_ (1302.70 g mol^−1^): *Calc.* C, 41.49; H, 2.79; N, 12.90. *Found:* C, 41.38; H, 2.69; N, 12.67%. Melting point: 332 °C (decomp.). ESI-MS: *m*/*z* = 691 [Tb(C_15_H_12_N_3_O_2_)_2_]^+^, 611 [TbC_15_H_12_N_3_O_2_(NO_3_)_3_]^−^, 487 [TbC_15_H_12_N_3_O_2_NO_3_]^+^, 268 [C_15_H_13_N_3_O_2_+H]^+^. IR (ATR, cm^−1^): ν = 3666 w (OH), 3057 w, 2975 w (N-CH_3_), 1637 s (C=N), 1608 m (C=C), 1460–1283 s, 828 m (NO_3_^−^), 1242 m (C–O), 1027 m (N–N) 506 w (Tb–O), 440 w (Tb–N) cm^−1^. UV-Vis (CH_3_CN): λ_max_/nm (ε/dm^3^ mol^−1^ cm^−1^) 307.2 (4.5 × 10^4^, 346.0 (3.1 × 10^4^).

***(4)*** 
*[Dy_2_(C_15_H_12_N_3_O_2_)_3_(NO_3_)_3_]·CH_3_CN **group A***


Light yellow precipitate, Yield: 18.67 mg, 0.0142 mmol, 53%. Analytical data Calculated C_45_H_36_N_12_O_15_Dy_2_ (1309.85 g mol^−1^): *Calc.* C, 41.26; H, 2.77; N, 12.83. *Found:* C, 41.06; H, 2.84; N, 12.61%. Melting point: 322 °C (decomp.). ESI-MS: *m*/*z* = 696 [Dy(C_15_H_12_N_3_O_2_)_2_]^+^, 616 [DyC_15_H_12_N_3_O_2_(NO_3_)_3_]^−^, 268 [C_15_H_13_N_3_O_2_+H]^+^. IR (ATR, cm^−1^): ν = 3665 w (OH), 3066 w, 2937 w (N-CH_3_), 1639 s (C=N), 1606 m (C=C), 1464–1289 s, 825 s (NO_3_^−^), 1249 m (C–O), 1035 m (N–N) 513 w (Dy–O), 438 w (Dy–N) cm^−1^. UV-Vis (CH_3_CN): λ_max_/nm (ε/dm^3^ mol^−1^ cm^−1^) 307.2 (3.7 × 10^4^).

***(5)*** 
*[Ho_2_(C_15_H_12_N_3_O_2_)_3_(NO_3_)_3_]·3CH_3_CN·CH_3_OH **group B***


Light pink precipitate, Yield: 25.28 mg, 0.0192 mmol, 64%. Analytical data Calculated for C_45_H_36_N_12_O_15_Ho_2_ (1314.71 g mol^−1^): C, 41.11; H, 2.76; N, 12.78. *Found:* C, 41.07; H, 2.69; N, 12.53%. Melting point: 267 °C (decomp.). ESI-MS: *m*/*z* = 697 [Ho(C_15_H_12_N_3_O_2_)_2_]^+^, 617 [HoC_15_H_12_N_3_O_2_(NO_3_)_3_]^−^, 493 [HoC_15_H_12_N_3_O_2_NO_3_]^+^, 268 [C_15_H_13_N_3_O_2_+H]^+^. IR (ATR, cm^−1^): ν = 3660 w (OH), 3060 w, 2962 w (N-CH_3_), 1644 s (C=N), 160 m7 (C=C), 1475–1285 s, 829 m (NO_3_^−^), 1245 m (C–O), 1028 m (N–N) 501 w (Ho–O), 441 w (Ho–N) cm^−1^. UV-Vis (CH_3_CN): λ_max_/nm (ε/dm^3^ mol^−1^ cm^−1^) 308.0 (4.4 × 104), 352.0 (2.8 × 10^4^).

***(6)*** 
*[Er_2_(C_15_H_12_N_3_O_2_)_3_(NO_3_)_3_]·CH_3_CN **group A***


Light yellow precipitate, Yield: 25,72 mg, 0.0193 mmol, 65%. Analytical data Calculated for C_45_H_36_N_12_O_15_Er_2_ (1319.37 g mol^−1^): *Calc.* C, 40.97; H, 2.75; N, 12.74. *Found:* C, 40.89; H, 2.81; N, 12.63%. Melting point: 310 °C (decomp.). ESI-MS: *m*/*z* = 700 [Er(C_15_H_12_N_3_O_2_)_2_]^+^, 557 [ErC_15_H_13_N_3_O_2_(NO_3_)_2_]^+^, 268 [C_15_H_13_N_3_O_2_+H]^+^. IR (ATR, cm^−1^): ν = 3677 m (OH), 3067 w, 2944 w (N-CH_3_), 1643 s (C=N), 1605 m (C=C), 1460–1283 s, 828 s (NO_3_^−^), 1251 m (C–O), 1037 m (N–N), 512 w (Er–O), 437 w (Er–N) cm^−1^. UV-Vis (CH_3_CN): λ_max_/nm (ε/dm^3^ mol^−1^ cm^−1^) 307.5 (3.7 × 10^4^).

***(7)*** 
*[Tm_2_(C_15_H_12_N_3_O_2_)_3_(NO_3_)_3_]·4CH_3_CN **group B***


Light yellow precipitate, Yield: 25.17 mg, 0.0190 mmol, 66%. Analytical data Calculated for C_45_H_36_N_12_O_15_Tm_2_ (1322.11 g mol^−1^): *Calc.* C, 40.86; H, 2.74; N, 12.71. *Found:* C, 41.08; H, 2.82; N, 12.69%. Melting point: 322 °C (decomp.). ESI-MS: *m*/*z* = 701 [Tm(C_15_H_12_N_3_O_2_)_2_]^+^, 620 [TmC_15_H_12_N_3_O_2_(NO_3_)_3_]^−^, 268 [C_15_H_13_N_3_O_2_+H]^+^. IR (ATR, cm^−1^): ν = 3680 m (OH), 3066 w, 2947 w (N-CH_3_), 1639 s (C=N), 1605 m (C=C), 1465–1288 s, 826 m (NO_3_^−^), 1249 m (C–O), 1037 m (N–N), 518 w (Tm–O), 438 w (Tm–N) cm^−1^. UV-Vis (CH_3_CN): λ_max_/nm (ε/dm^3^ mol^−1^ cm^−1^) 310.5 (4.2 × 10^4^).

***(8)*** 
*[Yb_2_(C_15_H_12_N_3_O_2_)_3_(NO_3_)_3_]·4CH_3_CN **group B***


Light yellow precipitate, Yield: 25.11 mg, 0.0188 mmol, 63%. Analytical data Calculated for C_45_H_36_N_12_O_15_Yb_2_ (1330.96 g mol^−1^): *Calc.* C, 40.61; H, 2.73, N, 12.63. *Found:* C, 40.81; H, 2.85; N, 12.72%. Melting point: 322 °C (decomp.). ESI-MS: *m*/*z* = 706 [Yb(C_15_H_12_N_3_O_2_)_2_]^+^, 626 [YbC_15_H_12_N_3_O_2_(NO_3_)_3_]^−^, 268 [C_15_H_13_N_3_O_2_+H]^+^. IR (ATR, cm^−1^): ν = 3663 w (OH), 3038 w, 2950 w (N-CH_3_), 1638 s (C=N), 1606 m (C=C), 1462–1286 s, 827 m (NO_3_^−^), 1247 m (C–O),1030 m (N–N), 519 w (Yb–O), 441 w (Yb–N) cm^−1^. UV-Vis (CH_3_CN): λ_max_/nm (ε/dm^3^ mol^−1^ cm^−1^) 307.0 (3.7 × 10^4^).

### 3.5. X-ray Crystallography

Diffraction data were collected by the ω-scan technique, for HL and 6 at 130(1) K, using mirror-monochromated CuKα radiation (λ = 1.54178 Å), on Rigaku SuperNova four-circle diffractometer with Atlas CCD detector, and in all other cases at 100(1) K with graphite-monochromated MoKα radiation (λ = 0.71073 Å), on Rigaku XCalibur four-circle diffractometer with EOS CCD detector. The data were corrected for Lorentz-polarization as well as for absorption effects [45]. Precise unit-cell parameters were determined by a least-squares fit of the reflections of the highest intensity, chosen from the whole experiment. The structures were solved with SHELXT [46] and refined with the full-matrix least-squares procedure on F2 by SHELXL [47]. All non-hydrogen atoms were refined anisotropically. All hydrogen atoms were placed in idealized positions and refined as ‘riding model’ with isotropic displacement parameters set at 1.2 (1.5 for CH_3_) times Ueq of appropriate carrier atoms. In the structure b one of the solvent molecules (methanol) was found disordered over two positions for which half occupancies were assigned. Some restraints were applied for these molecules (DFIX, ISOR). In **3a**, weak restraints for the displacement ellipsoids were also applied. Some of the crystals were of relatively poor quality, and the attempts to obtain the better ones failed. Therefore, in some cases, alerts appeared in the checking procedure [48]. However, the basic structural features are reasonably proved.

## 4. Conclusions

Reactions of lanthanides with polydentate oxygen and nitrogen ligands have attracted great interest because of the ability of these sites to realize stable chelate complexes with high coordination numbers. The Schiff base ligand **HL** and its corresponding lanthanide complexes of the composition [Ln_2_**L**_3_(NO_3_)_3_]·solvent (Ln = Sm^3+^ (**1**), Eu^3+^ (**2**), Tb^3+^ (**3a**, **3b**), Dy^3+^ (**4**), Ho^3+^ (**5**), Er^3+^ (**6**), Tm^3+^ (**7**), Yb^3+^ (**8**)), were synthesized and characterized. Their identity was further confirmed by the X-ray diffraction of single crystals. Interestingly, the details of the metal coordination patterns are different among the obtained complexes. These results have provided valuable information for possibility of designing new lanthanide complexes with structural chemistry depending upon crystallization conditions. The luminescent properties of ligand **HL** and compounds **1**–**8** have been also studied, both in solution and solid state. The efficient intramolecular energy transfer process from the triplet state energy level of ligand to excited energy level of lanthanide(III) ions is one of the factors influencing the luminescence properties of lanthanide(III) complexes. The occurrence of the luminescence of ligand **HL** and the absence or weak luminescence of characteristic emission bands of lanthanide ions in visible region is probably due to the large energy gap between the lowest triplet state level of the ligand and the excited state of lanthanides. The intense photoluminescence in the near-IR region was observed for Er(III) (**6**) and Yb(III) (**8**) complexes.

## Data Availability

Crystallographic data for the structural analysis has been deposited with the Cambridge Crystallographic Data Centre, Nos. CCDC-1542869 (HL), 2062743 (1), 2062744 (2), 1542870 (3a), 2062745 (3b), 2062746 (4), 1542871 (5), 2062747 (6), 2062748 (7) and 2062749 (8). Copies of this information may be obtained free of charge from: The Director, CCDC, 12 Union Road, Cambridge, CB2 1EZ, UK. Fax: +44(1223)336-033, e-mail: deposit@ccdc.cam.ac.uk, or www:www.ccdc.cam.ac.uk (accessed on 29 November 2022).

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
