# Peer review of "Synthesis and Characterization of Lanthanide Metal Ion Complexes of New Polydentate Hydrazone Schiff Base Ligand"

_molecules, 2022, doi:10.3390/molecules27238390_

Round 1

Reviewer 1 Report

My recommendation is to accept the manuscript in present form.

Author Response

We can only thank the Reviewer.

Reviewer 2 Report

The article deals with novel lanthanide complexes with polydentate hydrazone Shiff base ligand. The composition of new complexes was proved by X-Ray diffraction of single crystals, ESI-MS, CHN-microanalysis and IR-spectroscopy.
There are several major remarks to this work:
1. X-ray powder diffraction was not used, therefore there is no confidence that the samples are single-phase, and there is no evidence that the bulk sample consists of the same phase with single crystal.
2. The structure of the complex in solution and in the solid phase for labile lanthanide complexes does not necessarily coincide, and therefore the absorption and emission spectra may relate not only to those complexes that crystallize from the solution. In particular, spectrum 3b at first glance is a spectrum of a mixture of HL and 3a.
3. It is necessary to record the low-temperature phosphorescence spectra of the gadolinium complex with the same ligand to determine the triplet level of the ligand. Otherwise, the explanation of the absence or weak luminescence of characteristic emission bands of lanthanide ions in visible region looks insufficiently substantiated.
In addition, there are many other specific remarks:
1. It is necessary to add a picture with the structure of the ligand. At the first time the ligand appears as a crystal structure only on the third page, and it is shown schematically only in the SI;
2. It is also necessary to schematically depict the general structure of the resulting complexes to simplify the understanding of the text;
3. Table S1:
The unit cell parameters of the Tb(3b) structure do not correspond to cubic ones, while the specified symmetry group is cubic. In addition, in the table and text for this structure different groups are indicated - in table P213, and in the text (line 131) – Pa-3;
4. Figure S3 - mass loss curves do not reach a plateau;
5. Line 171 - what is the n-π* transition?
6. From line 197 the font changes;
7. Figure 5. What is the nature of the peak at 270 nm in spectrum 3b? Why does this peak not appear in spectrum 3a?
8. Figure 7. There is no explanation in the text for the fact that there are no lanthanide emission bands in the emission spectra;
9. Figure 8. Transitions to 1050 and 1300 nm are not signed. It looks like there are some transitions, what do they refer to?
10. Lines 235, 240 and etc. - it is necessary to make superscript and subscript indexes for terms;
11. For convenience, in the ciphers of complexes 1-8, it is worth indicating what kind of lanthanide is there, otherwise you constantly have to look at the table with structural data.

Author Response

Rev.2

The article deals with novel lanthanide complexes with polydentate hydrazone Shiff base ligand. The composition of new complexes was proved by X-Ray diffraction of single crystals, ESI-MS, CHN-microanalysis and IR-spectroscopy.
There are several major remarks to this work:

Dear Reviewer, thank you very much for you valuable opinion. We endorsed your contribution very much and decided to rewrite parts of the manuscript accordingly (it is marked in yellow in the revised text on).

1. X-ray powder diffraction was not used, therefore there is no confidence that the samples are single-phase, and there is no evidence that the bulk sample consists of the same phase with single crystal.

We performed X-ray powder diffraction studies (Fig.1; figures are in attached document), but they showed too much difference in the spectra. This was influenced by the Presence of a large number of solvent molecules in the structure of the monocrystals. Taking the single crystal out of the solvent phase and keeping it in laboratory conditions for a longer period of time causes it to become cloudy, which indicates the loss of solvents. This fact, in turn, affects the deformation of the planes.

  1. The structure of the complex in solution and in the solid phase for labile lanthanide complexes does not necessarily coincide, and therefore the absorption and emission spectra may relate not only to those complexes that crystallize from the solution. In particular, spectrum 3b at first glance is a spectrum of a mixture of HL and 3a.

We agree with the Reviewer's comment that the structures of the complexes in solution and in the solid may differ. However, in our opinion, the spectrum of complex 3b is not the spectrum of a mixture of ligand and complex 3a. In order to confirm our position, we deconvoluted the recorded UV-Vis spectra of the ligand and both compounds as shown in the attached figures (rev_2_fig_2).

 As can be seen from the spectra, a 383 nm band can be assigned for complex 3b, which is not present in either the ligand or 3a, clearly confirming its different structure.

  1. It is necessary to record the low-temperature phosphorescence spectra of the gadolinium complex with the same ligand to determine the triplet level of the ligand. Otherwise, the explanation of the absence or weak luminescence of characteristic emission bands of lanthanide ions in visible region looks insufficiently substantiated.

We agree with the Reviewer's comment that recording the low-temperature phosphorescence spectrum of the Gd(III) complex would allow us to determine the level of the triplet ligand. Unfortunately, at this moment we do not have this complex and cannot report the level of the triplet ligand. However, our studies have shown that weak luminescence of Eu(III) and Tb(III) ions is observed in solution while no luminescence associated with the presence of Sm(III) and Dy(III) ions was observed. This may indicate that the triplet state energy of the ligand is more suitable for the luminescence of Tb(III) and Eu(III) than Sm(III) and Dy(III). Furthermore, the broad band observed in solutions in the 400-500 nm range was attributed to the HL ligand emission in the study. A less intense one is observed in the Tb(III) and Eu(III) complexes (Fig.9a), indicating that energy transfer from the HL ligand to the metal in the Tb(III) and Eu(III) complexes takes place. Also, the observed higher emission intensity of Tb(III) ions in the complex, where coordination of both Tb(III) ions by the Schiff base (3a) takes place, confirms this fact.

We did not include the gadolinium ion in our study, as it is the subject of research by the co-authors of this paper. The results of the ligand HL studies with the gadolinium ion have been described and sent for review*.

* Marcinkowski, Dawid; Fik-JaskóÅ‚ka, Marta; Kubicki, Maciej; Patroniak, Violetta; Stefaniuk, Ireneusz; Wencka, M.; Slusarski, Tomasz; Piwowarska, Danuta; Gnutek , PaweÅ‚; Korabik, Maria; Karbowiak, Miroslaw; GorczyÅ„ski, Adam; Rudowicz, Czeslaw; Low and high symmetry aspects at the interface between the semiempirical and ab initio modelling of ZFS parameters and EMR spectroscopy – study case: trigonal and triclinic binuclear GdIII complexes, Dalton Transactions, Manuscript ID:DT-ART-08-2022-002847- under revision

In addition, there are many other specific remarks:
1. It is necessary to add a picture with the structure of the ligand. At the first time the ligand appears as a crystal structure only on the third page, and it is shown schematically only in the SI;

Thank you very much for this comment, the Scheme has been moved to the head manuscript: Scheme 2.

2. It is also necessary to schematically depict the general structure of the resulting complexes to simplify the understanding of the text;

The diagram is included in the SI as Scheme S1.

  1. Table S1:

The unit cell parameters of the Tb(3b) structure do not correspond to cubic ones, while the specified symmetry group is cubic. In addition, in the table and text for this structure different groups are indicated - in table P213, and in the text (line 131) – Pa-3;

Thank you very much for this comment; indeed it was our fault – the data are corrected, the space group is P213 in the text also. We have checked – for ssfety – all other data and they look ok.

  1. Figure S3 - mass loss curves do not reach a plateau;

TGA spectra were included to confirm the absence of solvents in the solid state. Examination of the melting points of the complexes showed decomposition of the complexes (only a color change to black in the solid Photo 1 (Dy(4)) was observed in all cases. This means that in the sample, in addition to lanthanide oxides, we have combustion products of organic fragments. (fig. 3)

  1. Line 171 - what is the n-π* transition?

The band with maximum at 331 nm, as shown in Fig. 5, can be assigned to n–p* transitions of conjugation between the lone pair of electrons of C=N groups and a conjugated p bond of the aromatic ring. Ref. [34-36]

We have changed the literature items number 35 and 36.

  1. From line 197 the font changes;

Thank you for this comment, the changes have been made.

  1. Figure 5. What is the nature of the peak at 270 nm in spectrum 3b?
    Why does this peak not appear in spectrum 3a?

The signal observed in the spectrum of compound 3b located at about 270 nm is not an apparatus artefact and this signal is only observed in the UV-Vis spectrum of this compound, as shown in the summary figure of the UV-Vis spectra of acetonitrile solutions of the studied complexes (Fig.S4). This is probably related to the presence of (C2H5)3NHNO3 in its structure, as reported by X-ray studies. In the spectra of alkyl nitrates, in low-polar solvents, there are weak absorption bands located at the aforementioned wavelength.

  1. Figure 7. There is no explanation in the text for the fact that there are no lanthanide emission bands in the emission spectra;

Thank you for this remark. We have added a suitable explanation to the main text:

„The very high intra ligand luminescence observed in the 400-650 nm spectral range in the solid compounds is indicative of non-efficient intramolecular energy transfer from the ligand to the excited states of Ln(III) ions and the lack of their bands in emission spectra.”

  1. Figure 8. Transitions to 1050 and 1300 nm are not signed. It looks like there are some transitions, what do they refer to?

Thank you for this comment. The corresponding transitions related to the observed bands have been inserted into the main manuscript:

„The low intensity signals observed at about 1050 and 1300 nm are attributed to the 5S2-5I6 and 5S2-5I5 transitions observed in this ion.”

  1. Lines 235, 240 and etc. - it is necessary to make superscript and subscript indexes for terms;

Thank you for this comment, the changes have been made.

  1. For convenience, in the ciphers of complexes 1-8, it is worth indicating what kind of lanthanide is there, otherwise you constantly have to look at the table with structural data.

Thank you for this comment, the changes have been made and made the text much easier to understand.

Reviewer 3 Report

The paper "Synthesis and characterization of lanthanide metal ion complexes of new polydentate hydrazone Schiff base ligand" is devoted to the sythesis and spectral study of eight complexes of lanthanides with hydrazone. Authors observed an interesting peculiarity of the obtained complexes: despite all of them have the composition of M2L3, the complex some cases is symmetrical, while in other complexes one metal bind all three ligands, while another is cordinated only by nitrates. The use of complementary methods is a bif advantage of the manuscript: for example, the bidentate coordination of nitrates observed from X-Ray data was additionally confirmed by IR spectroscopy.

There are only a couple minor questions:

1. It is known that, as you mentioned in Introduction, the coordination number (CN) of lanthanides varies from 8 to 9 and depends on the position of metal in the lanthanides raw: the starting elements (like La, Ce etc.) tend to have CN of 8, ending metals (Yb, Lu) - CN of 9, while the intermediate (Gd, Tb) can have either 8 to 9 [10.1007/s10953-020-00960-w]. You studied several lanthanides starting from Sm, which probably should has CN of 8 but it  hasn't. Do you have some suggestions why?

2. Is there some regularity in the dependence of type of complex (A or B) on the lanthanide number? Can the lanthanide tetrades somehow explain why A or B complex is formed?

3. The polydentate hydrazones in aqueous solutions tend to form the complexes of ML2 stoichiometry [10.1134/S0036023621100053]. Is it linked somehow with the type of complex (for example, complex type B being dissolved loses coordinating nitro groups and transforms to ML2 structure)?

Author Response

Rev. 3

The paper "Synthesis and characterization of lanthanide metal ion complexes of new polydentate hydrazone Schiff base ligand" is devoted to the sythesis and spectral study of eight complexes of lanthanides with hydrazone. Authors observed an interesting peculiarity of the obtained complexes: despite all of them have the composition of M2L3, the complex some cases is symmetrical, while in other complexes one metal bind all three ligands, while another is cordinated only by nitrates. The use of complementary methods is a bif advantage of the manuscript: for example, the bidentate coordination of nitrates observed from X-Ray data was additionally confirmed by IR spectroscopy.

There are only a couple minor questions:

 Dear Reviewer, thank you very much for your feedback and interesting questions .Below are the answers to the questions asked.

  1. It is known that, as you mentioned in Introduction, the coordination number (CN) of lanthanides varies from 8 to 9 and depends on the position of metal in the lanthanides raw: the starting elements (like La, Ce etc.) tend to have CN of 8, ending metals (Yb, Lu) - CN of 9, while the intermediate (Gd, Tb) can have either 8 to 9 [10.1007/s10953-020-00960-w]. You studied several lanthanides starting from Sm, which probably should has CN of 8 but it  hasn't. Do you have some suggestions why? 

Experiments related to the study of lanthanide complex [1,2] compounds clearly indicate that obtaining complexes with stoichiometric ratios of M:L 2:3, especially with tridentate chelate ligands, is common. The formed structure of the compounds is influenced by the applied stoichiometric ratio and the type of counter ions in the salt. The use of a stoichiometric ratio of salt : ligand equal to 2:3 in the synthesis was aimed at obtaining structures with interesting architecture, which was achieved. If a number of lanthanides are used for syntheses with the same ligand and in the same molar ratio, they mostly obtain the same coordination number in complexes.

1.Patroniak, V., Stefankiewicz, A. R., Lehn, J.-M., Kubicki, M., & Hoffmann, M. (2006). Self-Assembly and Characterization of Homo- and Heterodinuclear Complexes of Zinc(II) and Lanthanide(III) Ions with a Tridentate Schiff-Base Ligand. European Journal of Inorganic Chemistry, 2006(1), 144–149. doi:10.1002/ejic.200500699

  1. GorczyÅ„ski, A., Kubicki, M., Pinkowicz, D., PeÅ‚ka, R., Patroniak, V., & Podgajny, R. (2015). The first example of erbium triple-stranded helicates displaying SMM behaviour. Dalton Transactions, 44(38), 16833–16839. doi:10.1039/c5dt02554k
  2. Is there some regularity in the dependence of type of complex (A or B) on the lanthanide number? Can the lanthanide tetrades somehow explain why A or B complex is formed? 

Analysis of the results obtained for complexes 1-8, suggests that the obtaining of the structure from group A or B was influenced by the crystallization conditions, especially the type of solvent in the external phase.

  1. The polydentate hydrazones in aqueous solutions tend to form the complexes of ML2 stoichiometry [10.1134/S0036023621100053]. Is it linked somehow with the type of complex (for example, complex type B being dissolved loses coordinating nitro groups and transforms to ML2 structure)? 

In fact, we did not aim at performing experiments in aqueous solution, so this was not our concern. Anyway, while analyzing the ESI-MS spectra of the complexes, for both group A and B performed in methanol, we observed the presence of fragmentation ions (in the harsh conditions of the measurements), indicating the presence of nitrate ions in the coordination sphere of the central ion.